# Planet YouPorn: Pornography, Worlding, and Banal Globalization in Michel Houellebecq's Work

**Gustaf Marcus**

Department of Modern Languages, Uppsala University, 751 26 Uppsala, Sweden; gustaf.marcus@moderna.uu.se

**Abstract:** This article studies mediated erotic content, especially pornography, as a form of worlding in Michel Houellebecq's work. Whereas love creates a space of alterity, pornography paradoxically combines the most intimate spatiality of the body with ever-expanding technological systems and global forms of mediation. This short-circuiting of space points to a new sense of being in the world, which is studied in selected passages from the novels *La Possibilité d'une île* and *Soumission*, as well as in the essay "Prise de contrôle sur Numéris." With reference to Ulrich Beck's description of "banal cosmopolitanism," I argue that otherness is either reduced to free-floating objects of consumption or to an experience of absence in these texts. Furthermore, this duality is refracted as two "reflexively" interwoven discourses or voices in the work. One is associated with prose and with the bringing of the world to the body of the subject, and the other with poetry and the dissolution of the body into the space of the world.

**Keywords:** Michel Houellebecq; pornography; globalization; worlding; Ulrich Beck; Anthony Giddens; banal cosmopolitanism; reflexivity





> *Ce qui compte c'est d'être au monde, peu importe la posture, du moment qu'on est sur terre.*—Samuel Beckett, *Textes pour rien*

## 1. Love Creates a World

Michel Houellebecq is an author for whom the nature of the "world" and the experience of being part of a world are central questions. This has been highlighted several times. The novels revolve around our representations of the "space of the world," as Daniel Laforest puts it, and the cultural practices (such as charter tourism) that make these representations possible (Laforest 2007, p. 265). Sylvain Bourmeau, to mention another example, identifies what he calls a "Google Earth-effect" in the work, a formulation that neatly captures its combination of sweeping global scope and attention to minute details (Bourmeau 2017, p. 191). There is also something like a distinctive "attitude" toward the world or "posture," to use Beckett's formulation from the epigraph of this article; we might not be able to change the world, but the question of how we *are in* or *relate to* the world is constantly at the heart of Houellebecq's work.

In this article, I use the theme of love, as well as its technological and globalized counterpart, mass-mediated pornography, as a point of departure for exploring these questions. My goal is not only to elucidate these particular themes, but also to provide a theoretical framework, which has been lacking in the earlier studies, for understanding the question of "worlding" more broadly in the work. This theoretical framework is provided by Ulrich Beck's concept of "banal cosmopolitanism" (Beck 2004, 2006) and Beck's and Anthony Gidden's accounts of "reflexivity" (Beck et al. 1994; Giddens 1990, 1991).

Love is my starting point, because it is a utopic attachment that makes it possible to transcend our own finitude in time and space in Houellebecq's work. In this sense, love creates a world. In one of the love poems at the end of *Le Sens du combat* (*The Art of Struggle*, 1996), the lover finds the key to new "world structures" (*structurations du monde*) in the beloved person, and the poem ends with the triumphant: "And there is a new

world." (Houellebecq 2014a, p. 132). Such formulations appear with insistent regularity. In Sérotonine (2019), the narrator claims that a woman's love is "the source of [...] a different universe" and that "[t]hrough love, women create a new world" (Houellebecq [2019] 2020, p. 58). Sex, in particular, is often described as a transcending of the self and a merging with the space of the world. The eroticized encounter between the volcano and the sea creates what appears as a new world in one novel (*Lanzarote* (Houellebecq 2000a)). And in another one (*Les Particules élémentaires* (*The Elementary Particles*, 1998)), swingers having sex on a beach in Cap d'Agde seemingly turn into cells in a vast nervous system as the sand dunes become folds in a giant brain. In the same novel, "love" turns out to be another word for complex quantum states that unite people and the universe, and the hero scientist Michel modifies the DNA of human beings to be more attuned to this mysterious spatiotemporality. It ends with the creation of an interconnected body-world: "a perceived world, a felt world, a human world" (Houellebecq [1998] 2000b, p. 249).

It is often the *space* between the self and the other that is experienced as distance *and* as presence. Space turns into experiential place through the mysterious "human presence" that Houellebecq often alludes to.[1] As Michel puts it in *Les Particules*, "the lover hears his beloved's voice over mountains and oceans" in a way that creates a "a magnificent interweaving" (Houellebecq [1998] 2000b, p. 251). My point is that this "interweaving" contains both proximity and geographical distance, both the self and the other. The lovers' bodies are like two parallel mirrors: though a single mirror "day after day [...] only returns the same desperate image," two mirrors "elaborate and edify a clear and dense system," a space that is "infinite in its geometrical purity, beyond all suffering and beyond the world" (Houellebecq [1994] 2011, p. 146). It is the "world between the skins" that emerges between the lovers, as in the poem displayed in "La Rivière," the erotic short film that Houellebecq wrote and directed for Canal+ in 2001:

LE MONDE ENTRE LES PEAUX

LA DOUCEUR DES SECONDES

INSTALLE UN NOUVEAU MONDE

[...]

NOUS RECRÉONS UN MONDE

ENLACÉ DE CARESSES

UN MONDE ENTRELACÉ

NOUS RECRÉONS L'ESPACE

NOUS RECRÉONS L'ESPÈCE. (Houellebecq 2001a, 2:30–14:43)

(THE WORLD BETWEEN THE SKINS/THE SOFTNESS OF THE SECONDS/ESTABLIS HES A NEW WORLD/[...] WE RECREATE A WORLD/INTERLACED WITH CARESSES/AN INTERLACED WORLD/WE RECREATE SPACE/WE RECREATE THE RACE.)

Time, space, self, and other all turn into a great "interweaving" through the movement of the repetitions, the verse, and the rhymes: *secondes/monde*; *espace/espèce* (*seconds/world*; *space/race*). Moreover, this interlacing is not only poetic; it is what poetry as an inherently worlding force *does*. In fact, in several essays and interviews, Houellebecq describes poetry as the expression of a mode of perception in which distances and differences are effaced. It is a way of "seeing the world" that makes "all the objects in the world (highways as much as snakes, parking lots as much as flowers)" appear as aspects of a greater whole (Houellebecq 2022, p. 40).[2]

However, as the reader quickly notices, the "time-space compression" of modernity, with its ever-expanding technological systems and forms of mediation, complicate this idealized copresence of the lovers in a "human" space. If love creates a world, what about pornography? Or other mediated forms of romantic love and erotic exchange over potentially great distances, such as escort sites, dating apps, audiovisual performances, or

even the old Minitel's 3615 Aline and *messageries roses* (all motifs that figure prominently in the work)? These questions are unsettling because they highlight Houellebecq's own ambivalent relationship to pornography. Even an ordinary pornographic website has many similarities with the great web of interrelations that reshapes time and space in *Les Particules*. Here too, connections are established that defy time and space. For example, the hundreds of thousands of videos on the video-sharing website YouPorn are, as the deeply impressed François in *Soumission* (*Submission*, 2015) notes, produced in every corner of the world (Houellebecq [2015] 2016, p. 17). According to the website's own data, around 250,000 videos are uploaded each year, attracting more than 5 billion annual visits from around the globe (YouPorn 2019). Put together, the internet platform constitutes an interactive and multifaceted image of a "human world" that is completely globalized, yet easily accessible at François' fingertips in his Paris apartment. But the immediacy of access erodes the experience of alterity, and the high-technological cyberspace challenges the idea of a "world."

We could perhaps modify the passage from *Les Particules* about the lover who hears the call from the beloved "over mountains and oceans" to be better suited for this situation: "The isolated male hears the moaning of the pornstar over mountains and oceans, in HD and on-demand." But now the question becomes pressing: what happens with the experience of alterity—the copresence of self and other and of proximity and distance—in this new "space"?

## 2. Structure of the Article

This question will guide my reflection in this article. In the first part, I present the theme of pornography more generally in Houellebecq's work. Whereas technology is essentially a deterritorializing force, sexuality reinscribes the self and the body in the world. Pornography, which unites technology and sexuality, thus opens up an ambiguous space: what kind of "copresence" does technological globalization open up for the human animal? In order to answer this question, I take a closer look at the representations of pornography and especially at the connection between pornography and worlding through close readings of passages from the two novels *La Possibilité d'une île* (*The Possibility of an Island*, 2005) and *Soumission*, and of the essay or short story "Prise de contrôle sur Numéris" (Taking Control on Numéris, 1995).

I show that vast spaces and distances are systematically confronted with the intimate space of the body in these texts. At the same time, any intermediary space, or space of belonging *between* the isolated and eroticized body and the vast space of the globe, is eroded. This process, in turn, is mirrored in the zoom effects and use of iterative narration that characterizes Houellebecq's prose. More specifically, I compare this spatial tension in the pornographic scenes to a vertiginous "fall" toward the human and a "dolly zoom," or the unsettling cinematic effect that is created when a camera is zooming in on an object that it is simultaneously being pulled away from.

Then, in the second part of the article, I turn to theory in order to tease out a more general meaning from the close readings. I discuss Houellebecq's representations of pornography as expressions of "banal" globalization and what Beck and Giddens call "reflexivity" (Beck et al. 1994). Beck and Giddens systematically examine how deterritorialized expert systems, communication technologies, and technological risks create an ever-increasing "intersection of presence and absence" (Giddens 1991, p. 22) in late modernity. These global contexts that infiltrate the local lifeworlds are essentially unknowable for the individual, and the primordial division between "near" and "far away" is eroded (cf. Bauman 1998, pp. 13–15).

Similarly, I argue that alterity or difference is either eliminated through acts of detached or "banal" consumption in Houellebecq's work, or it is presented as an experience of lack or absence. Though the first "attitude" toward the world is the most common (and it is consistently associated with pornography), there is also a counter discourse in Houellebecq's work. In the final section of the article, I outline this other discourse that

reverses the relationship between the body and world: rather than prose, it is associated with poetry, and rather than bringing the world to the body of the subject, it points to the dissolution of the subject into the space of the world. These two voices or discourses mirror each other, and they are, I argue, complimentary responses to the experience of banal globalization.

### 3. *La Possibilité d'une île* and the "Dolly Zoom-Effect" of Pornography

Despite a rich secondary literature on pornography and sexuality, there is little agreement about the meaning of these prevalent themes in Houellebecq's work. The debate has often been moralizing, and it has largely revolved around the pornographic nature of the literary texts themselves, what James Dutton calls the "'Houellebecq, pornographer' trope" (Dutton 2022, p. 441).[3] Some scholars argue that Houellebecq's prolific use of pornographic stereotypes is another form of exploitation, a more or less direct continuation of the sex industry itself. Others contend that it should be seen as a strategy for demasking or criticizing the incessant commodification of sex in late capitalism. Mads Anders Baggesgaard, for example, convincingly argues that mediated erotic content introduces a visual distance that erodes the intimate and tactile experience of sexuality (Baggesgaard 2007). A third group, finally, identifies an ambivalent search for a middle ground or a dialectical tension between these positions.[4]

The literature on sex and sexuality in the work is almost too vast to summarize, but a particularly interesting group of studies focus on the importance of pornography as a *technological* challenge to human nature in the era of mass-mediation and gene technology. It is, however, still a point of contention whether sexuality in Houellebecq's work should be understood as essentially malleable and unstable, or as a primordial and unchanging force of nature. For example, Ben Jeffery laments that Houellebecq is "astonishingly naïve on the subject of sex, as though the pornographic is the only sort of fantasy he is unable to expose" (Jeffery 2011, p. 13). Douglas Morrey, on the contrary, argues that the theme of sexuality constitutes "a sustained reflection on our species' unstable position along a continuum that runs from the animal, through the human, to the posthuman" (Morrey 2013, p. 10). I will argue that both are paradoxically correct, using the posthuman novel *La Possibilité d'une île* as an example. As I will try to show, pornography and other forms of mediated erotic content are crucially on both ends of the spectrum that separates biology from technology as they inscribe the human body and human copresence in a world of technological flux and banal globalization.

The strange resilience of human sexuality (that Jeffery calls "naïve") is, in fact, one of the major themes in the novel. The comedian and film producer Daniel builds a brilliant career by relativizing and satirizing everything, from social inequalities, to parricide, and even torture. His postmodern *détournements* manages to turn everything into "laughter" and "money" (Houellebecq [2005] 2006, p. 134). But human sexuality is somehow different. As he is forced to admit, the basic "human" meaning of pornography remains unchanged despite all the attempts to mediate it differently: "The thing seemed up till now to resist all attempts at sophistication. Neither the virtuosity of the camera movements, nor the refinement of the lighting brought about the slightest improvement" (p. 134). However, this biological grounding crucially works both ways; what could be called the essential humanity of sexuality makes it possible to constantly deterritorialize it and inscribe it in new technological systems and forms of mediation. This paradox is clearly stated by another Houellebecquian protagonist, Florent-Claude in *Sérotonine*, in a reflection about YouPorn. As he puts it, "porn has always been at the at the forefront of technological innovation." This insight is shared by "numerous essayists," but they all fail to recognize what is so "paradoxical" about it, namely that

> pornography remains the sector of human activity in which there is least room for innovation, and absolutely nothing new happens; everything imaginable, broadly speaking, existed in the pornography of Greek or Roman antiquity. (Houellebecq [2019] 2020, pp. 94–95)

Sexuality is paradoxically linked to the most intimate instincts and functions of the body, while also being possible to inscribe in ever-expanding technological systems and forms of mediation. Even if it turns into pixels, numbers, or images, sexuality translates into a sense of "human presence," however alienated.

This tension comes to the fore when the *technological* clone at the end of *La Possibilité* (Daniel's clone who lives thousands of years in the future), who has lost almost every trace of his *biological* humanity, decides to leave his secluded life to try to track down Marie23. The Marie clone has sent him strange erotic messages, for example, a pixelated picture of her vagina, symbolically placed in the opening of the novel: "2711, 325104, 13375317, 452626. At the address indicated I was shown an image of her pussy—jerky, pixelated, but strangely *real*" (Houellebecq [2005] 2006, p. 4). The abrupt shift from the abstract IP-address to the intimate body part seems to compress distance and presence, just like the distorted erotic image combines the extremes of the technological and the biological. At first, the detached and asexual clone makes fun of the pixelated body part, this "pussy connected to mysteries—as though it were a tunnel opening on to the essence of the world" (p. 4). But this *worlding* force of sexuality soon sucks him in. Like a trapped animal, he can only regret his fatal curiosity in retrospect: "I should have stopped. Stopped the game, the intermediation, the contact; but it was too late" (p. 4).

Such calls "over mountains and oceans" (Marie's residence is literally on the other side of the Atlantic Ocean, among the ruins of what used to be New York) finally makes the Daniel clone leave his home for the first time as he travels a devastated post-apocalyptic world by foot in search of a remaining "neo-human" community on Lanzarote. By accident, he visits some of the places that his human predecessor, the original Daniel, had visited during his tumultuous life, but they are now devastated and scarcely populated after centuries of nuclear war and natural disasters. With the words from one of the poems in *Le Sens du combat*, the clone "falls toward the human" (Houellebecq 2014a, p. 133).

In fact, immediately after the image of Marie's vagina in the opening of the novel, a satellite image of Almeria appears on the clone's computer screen as if to hint at a fall toward the world of organic life. This is the region in Spain where his human predecessor spent the last part of his life. The clone's "fall" toward the human and the possibility of a merging of the two perspectives, the clone's vertical and technologically mediated perspective and the human ancestor's horizontal perspective, then becomes a focal point of the narrative. Especially the final section of the novel is a poetic meditation on the relationship between the human and the posthuman, on how human affects and emotions remain as a vague and distorted echo within a technologically altered body and in a devastated world.

This is a centrally important theme in Houellebecq's work in general. The narrators in *Sérotonine* and *Extension du domaine de la lutte* (*Whatever*, 1994) have similar dreams of falling toward the world from their alienated lives in Paris. Florent-Claude dreams about falling from the Tour Gan, the skyscraper in which he lives (Houellebecq [2019] 2020, p. 65), and the narrator in *Extension* from the towers of the Chartres Cathedral (Houellebecq [1994] 2011, pp. 141–43). Michel in *Les Particules* similarly falls to his death in the Atlantic Ocean. Furthermore, this "fall toward the human" is often, as in *La Possibilité*, associated with an ambivalent (re)awakening of the protagonists' sexuality. The clock tower in *Extension* is, for example, clearly a phallic symbol that is associated with the protagonist's ambivalent sexuality and fears of castration. Jed, the artist in *La Carte et le Territoire* (*The Map and the Territory*, 2010) who reproduces maps and satellite photographs, "falls" into the messy world of human sexuality when he meets the seductive Olga and subsequently leaves his Paris studio (placed in the attic of a tall building) to travel the French countryside (Houellebecq 2010). In fact, even Houellebecq's photographs are often taken from above in an oblique angle, which erases the line of the horizon from the picture.[5] Similar to the dolly zoom, this photographic technique produces a kind of vertigo effect, since it makes it hard for the viewer to determine the dimensions of the depicted landscapes without the line of the horizon as a stable reference point. Rock formations, sand dunes, and indistinct

vegetation become ambiguous in scale; they could be small and photographed from a close distance, or, just as well, massive and photographed from a great height—in one of them, "Inscriptions #015," the viewer even finds the provocative inscription "The world is medium-sized."

## 4. The "Normal World"

It is in this paradoxical, vertiginous space that the modern world takes shape. At one point in the *La Possibilité*, the original Daniel and the artist Vincent make an excursion into what Daniel repeatedly calls the "normal world" (Houellebecq [2005] 2006, pp. 224, 226) as he walks on a beach in Lanzarote. The two men are surrounded by tourists, half-naked women, who share the same physical space as them but who, like *ob-jects* in a supermarket or on a screen, are cut off from any shared context or locality and are simply "thrown" (lat. *jacere*) in front of them: "Our mutual sharing of space [*appartenance à un espace commun*] with them was fated to remain purely theoretical [. . . ] in our eyes they had no more existence than if they had been images on a cinema screen" (p. 224; Houellebecq [2005] 2012, p. 242). They are objectified accordingly as "teenage girls with attractive bodies" or as "mothers whose bodies were already less attractive" (Houellebecq [2005] 2006, p. 224). Of course, this is not pornography strictly speaking, but the beach functions, as Daniel points out, like a "screen" where the tourists' "mutual sharing of space" is tenuous and arbitrary. The most attractive women also congregate to make an erotic display in the middle of the beach, a "Miss Bikini Contest." And it is in this deterritorialized space that the "normal world" manifests itself as soon as the first girl steps on the stage:

> She came from Budapest. *"Budaaaa-pest! That city's hoooot!. . ."* [. . . ] He moved to the next girl, a platinum-blond Russian, very curvaceous in spite of her fourteen years [. . . ]. [. . . ] And so that was it: we were in the *normal world*. (pp. 225–26)

This association, in the words of the announcer, between detached erotic consumption and the space of the world ("*Budaaaa-pest! That city's hoooot!*") could perhaps seem inconsequential, but we have already seen how important the idea of a sexualized body-world is for Houellebecq. The passage confronts us with the same pressing questions that I alluded to in the introduction: how does global modernity shape spaces of copresence and alterity? And how do other people appear in this man-made "normal world"? As we will see, this connection between banal erotic consumption and figurations of the world returns with curious consistency throughout the work. Next, I will analyze two other examples of the body-world of pornography, before turning to globalization theory. First, I will comment on the early essay or short story "Prise de contrôle sur Numéris," which was originally published in the collective work *Objet perdu* (Alexandre et al. 1995), and then I will analyze François' detailed description of YouPorn in the 2015 novel *Soumission*.

## 5. Taking Control on Numéris

In what is possibly the first instance of pornography in Houellebecq's work, the narrator in "Prise de contrôle sur Numéris" masturbates as he watches pixelated "pornographic icons"—"faces of women moving up and down, sucking off male sex organs" (Houellebecq 1998a, p. 34)—on his early 90s computer. He also uses and closely describes the sex chatting and dating service 3615 Aline, which was created for the Minitel network at the end of the 1980s. And just like in the previously discussed passage in *La Possibilité*, spatiality immediately takes center stage in the essay as the prose description of the narrator's exploration of 3615 Aline is broken up by a poem that outlines a map of the globalized world:

> *L'Afrique sombre dans la mort*
> *Et les Polonais,*
> *Les pauvres Polonais,*
> *Semblent destinés une fois de plus à jouer le rôle de guignols du libre-échange.*
> *Pendant ce temps, l'Europe occidentale bascule dans le camp des pays moyen-pauvres;*
> *Situation "à la libanaise."* (p. 31)

*(Africa is going under / And the Poles, / The poor Poles, / Seem destined to become, once more, the puppets of free trade. / At the same time, Western Europe falls into the lower-middle income category; A "Lebanese" situation.)*

The poem describes what could be called the global wasteland of capitalism from a bird's eye view. In the prose part of the essay, the narrator explores this emerging spatiality from another point of view as he describes how contact can be established between people and how the world appears to the users of 3615 Aline. He partakes in the fight for sexual partners that is waged on the platform through the bits of information that are passed around at the speed of light, channeled through "fiber optic cables" and "multiplexing data centers" (p. 33). As the title suggests, this is an attempt to "take control" of a territory. But this is a paradoxical territory where no one is present in the sense of being situated, yet everyone is present everywhere. It is a territory of indeterminate size where every locality or node is interchangeable (though the Numéris network was still, strictly speaking, limited to France, the narrator mentions Al Gore's plans to develop an even more performant network connecting Europe to the American continent).

How do people appear in this networked environment? At one point, SANDRA.W introduces herself with the following message "165 58K 90TP" (p. 33). After a while, the narrator deduces the meaning: "Sandra" is 165 cm tall, weighs 58 kg, and has a bust size of 90 cm. The platform only allows the users four lines and 40 characters to present themselves, which leads to an explosion of inventiveness. Even though a haiku would probably fit in this space, the narrator notes that lyrical poetry is out of place in this environment as a description of beauty. It is obviously too vague and subjective to provide the basis for an "informed choice" (p. 33). And more fundamentally, *poiesis*, as a performative relationship between the subject and the object, presupposes presence in a shared space.[6] Instead, he suggests that the "progressive quantification of the world" requires "standardized sexual descriptions" (p. 35) that better suit the mode of appearance of individuals in this emerging networked territory. Inspired by the social security number (and SANDRA.W's profile), he promptly creates a twelve-digit number for describing men and a fourteen-digit number for describing women by adding the age of the person to a choice of biometrical data, for example: "159173651704, 26116144875585, 25516452925788" (p. 35).

The long lines of numbers in the text resemble IP addresses that encode localities in an abstract space, much like in the initial message that Marie sends to the Daniel clone in *La Possibilité*: "2711, 325104, 13375317, 452626. At the address indicated I was shown an image of her pussy." Even the pixelated sexual organs are present in both texts (in Marie's message and in the pornographic icons in the essay). Just like the novel, finally, "Prise de contrôle sur Numéris" explores the tension between the technological and the biological in late modernity that manifests itself as an indefinite expansion of space and a simultaneous reduction of experiential place. When the former is stretched out to encompass the globe, the latter is shrunk to the limits of the isolated body. This development is distinctly claustrophobic, but Houellebecq introduces an ironic or critical distance in the essay as the narrator's bleak verdict about poetry being supplanted by quantitative information is immediately contrasted with a *poem* about this quantification process:

> *La société est cela qui établit des différences*
> *Et des procédures de contrôle* [...]
> [...]
> *Le prix des choses et des êtres se définit par consensus transparent*
> *Où interviennent les dents,*
> *La peau et les organes,*
> *La beauté qui se fane* [...]. (Houellebecq 1998a, p. 33)

*(Society is that which creates differences / And control procedures* [...] / [...] *The pricing of things and beings are fixed by transparent consensus / In which teeth, / Skin, and organs count / The beauty that withers* [...].)

**6. Planet YouPorn**

Turning to contemporary and truly globalized networked spaces, such as the dating app Tinder or the pornographic site YouPorn, the amount of information in terms of megabytes that are being exchanged is vastly different, but the fundamental relationship between the desiring individual and the space of the world remains the same. Twenty years after "Prise de contrôle sur Numéris" was published, YouPorn is described in *Soumission* as a reflection of the "fantasies of normal men spread out over the entire planet [*répartis à la surface de la planète*]" (Houellebecq [2015] 2016, p. 17; 2015, p. 25). In other words, the site represents typical sexual fantasies *and* the space of the world. And, just like in the essay, François' description emphasizes how the space of the world becomes rearranged and thus possible to experience in new ways in relation to these mediated fantasies. Here is the passage that describes François' surfing on the pornographic site in its entirety:

> Some of the videos were superb (shot by a crew from Los Angeles, complete with a lighting designer, cameramen and cinematographer), some were wretched but "vintage" (German amateurs), and all were based on the same few identical and pleasing scenarios. In one of the most common, some man (young? old? both versions existed) had been foolish enough to let his penis curl up for a nap in his pants or boxers. Two young women, of varying race, alerted him to the oversight and, this accomplished, stopped at nothing until they had liberated his organ from its temporary abode. They coaxed it out with the sluttiest kind of badinage, all in a spirit of friendship and feminine complicity. The penis passed from one mouth to the other, tongues crossing paths like restless swallows in the sombre skies above the Seine-et-Marne, when they prepare to leave Europe for their winter pilgrimage. The man, destroyed at the moment of his assumption, only uttered a few weak words: appallingly weak in the French films ("*Oh putain!*" "*Oh putain je jouis!*": more or less what you would expect from a nation of regicides), more beautiful and intense from those true believers the Americans ("Oh my God!" "Oh Jesus Christ!"), like an injunction not to neglect God's gifts (blow jobs, roast chicken). At any rate I got a hard-on too, sitting in front of my twenty-seven-inch iMac, and all was well. (Houellebecq [2015] 2016, pp. 17–18; 2015, p. 26. I have slightly altered the translation.)

The paradoxical spatial movement in this elaborate description of an archetypical fellatio scene mirrors the cinematic dolly zoom effect. When a camera is zooming in while being pulled away, the focalized object (in this case, the two women) remains unchanged, but the size and sharpness of the foreground and background change around it. This technique creates a vertigo effect that makes the viewers lose their sense of space and orientation. In the cited passage, short-circuited connections between places and cultures are similarly established "around" the unchanging pornographic scene and the immediate bodily identification it produces in the narrator. The entire world is effectively deterritorialized, reordered, and finally represented in a burlesque fashion with the help of his iMac. To the isolated and placeless observer (even though François' name would indicate "frenchness"), the decomposed world manifests itself anew as the source of minimal differences clustered around the pornographic scene. He contrasts German amateur videos with professional American studio productions; the minimal linguistic differences are traced back to historical events, such as the French revolution and the founding of America by devout pilgrims. The world is first blurred, then it crystalizes into a new image.

The effect is created by the iterative narration, where something that happens many times in different locales is narrated once. This technique is common in Houellebecq's novels and he often pushes it over the limit into what Gérard Genette calls the "pseudo-iterative," where the narration of something that happens several times becomes so detailed that it seems singular (cf. Genette 1983, pp. 113–27).[7] The result is that the world seen through YouPorn appears compressed as a burlesque *image* that is structured according to deterritorialized acts of consumption, instead of being a *setting* that would be able to give meaning to François' actions.

The only way out of this claustrophobic, yet global space is seemingly the enigmatic landscape, the dark Seine-et-Marne sky, which opens up like an absurd poetic outgrowth when François compares the movement of the women's tongues to migratory birds. They are like "restless swallows in the sombre skies above the Seine-et-Marne, when they prepare to leave Europe for their winter pilgrimage." Here, the change in register is matched by the abrupt leap from mass-mediated pornography to the idea of a devout pilgrimage. But this is not merely an example of Houellebecq's famous use of *asyndeton*, or abrupt changes of style and register (cf. Viard 2008, pp. 55–60); it also emphasizes the spatial push-and-pull effect that characterizes the entire passage. The landscape that is geographically the *closest* to the Parisian narrator, the vast Île-de-France plain, is the one that seems most inaccessible and foreign when compared to the burlesque references to German and American pornographic videos. The inhuman movements of the migratory birds, finally, the swallows, which are capable of travelling across the globe in a matter of days, stand in sharp contrast to the stationary observer in front of his computer screen. "[T]here is no lesson to be learned from swallows," as Houellebecq puts it in an earlier prose poem, since they travel the "globe" while humans remain on Earth: "Swallows fly off, slowly skim the waves, and spiral up into the mild atmosphere; they do not speak to humans, for humans remain tied to the Earth." (Houellebecq 2017, p. 119). Squeezed between the contradictory movements of globalization and the loss of experiential place, locality is inscribed as absence in this vision of the world.

## 7. "We Inhabit Absence": Banal Globalization and Reflexivity

So far, I have argued that pornography combines the deterritorializing aspects of late modernity with sexuality as an essentially grounding force that reinstates a bodily presence in the world. This double movement manifests itself as a vertiginous "fall" toward the world or even a "dolly zoom," where the relationship between the subject and the object warps and reshapes the experience of the space of the world. In this section, the goal is to situate Houellebecq's texts in a wider theoretical perspective in order to tease out a more general pattern and experience of "worlding" in the work.

The peculiar push-and-pull effect explored in the commented texts on pornography corresponds to the phenomenological experience of "banal" globalization. I borrow this term from Beck, who uses it to highlight how the world of especially Western consumers has become *unintentionally* globalized, as a side effect or as the unreflective and automated background structure of our actions. "Banal cosmopolitanism," as Beck puts it, is a process in which the other is simultaneously radically present and radically absent.[8] Think, for example, of the papayas or mangos on display in a regular supermarket; they seem immediately present and accessible, yet they are shipped from far-away places and produced under conditions that are unknown and even (as Beck argues) *unknowable* for the consumer. They are, in this sense, like small irruptions of absence, much like the names of the different types of hummus that punctuate the flow of the text in *Sérotonine* when the narrator visits a Carrefour supermarket: "[t]he oriental food shelf [...] displayed [...] the abugosh premium, the misadot, the zaatar and the [...] mesabecha" (Houellebecq [2019] 2020, p. 76).[9] In the eyes of the banal consumer, the foreign objects are simply *there* without any intermediary preparation, as manifestations of a different spatio-temporal reality (as in one of the poems: "The calm of the objects [...] is strange/[...] Time cuts us to pieces but nothing causes them to change" (Houellebecq 2014a, p. 259)). And this abrupt intrusion of distance is emphasized by the unmediated use of foreign words and trademarks—another trait that is characteristic of Houellebecq's writing.[10]

According to Beck, such irruptions of absence are, in fact, omnipresent in contemporary post-industrial societies. Everything, from food to clothes and cellphones, depend on globalized regulations, forms of knowledge, and processes that become increasingly impenetrable. "Without my knowing or explicitly willing it, my existence, my body, my 'own life' become part of another world, of foreign cultures, religions and histories" (Beck 2004, p. 134). It is important to note that this "latent," or "passive" cosmopolitanism complicates

or even negates what Beck calls the traditional "normative" visions of cosmopolitanism, in which distinct cultural spheres are understood as merging or developing in relation to each other (some recurrent tropes in cultural studies that are used to describe this kind of relationship, whether it is repressive or friendly, are *travel*, *circulation*, *translation*, *exchange*, *hybridity*, *mestizaje*, and *vernacularity*).[11] Translated into Houellebecq's language, this "traditional" model of globalization and cultural exchange would be similar to the utopic discourse of love outlined in the introduction: the self and the other meet in the space that separates the two, and they are transformed through an encounter that, in turn, engenders a world. But this is where the banal globalization of pornography, where experiential place and the experience of alterity are radically challenged, provides a model for a different, darkly dialectical form of worlding that is impossible to grasp with those terms.

To understand this process, it will instead be useful to revisit another one of Beck's central theoretical concepts: reflexivity (cf. Beck et al. 1994). In the globalized world, many sectors of everyday life, such as the economy, media, or global institutions dealing with public safety, are working on a local level, but their functioning and "built-up" are truly global. They rely on globalized knowledge production, globally produced and dispersed technologies, and global risk scenarios and forecasts. Here, datamining and deterritorialized expert systems feed back seamlessly and with increasing rapidity into everyday life until the whole cloth out of which the local lifeworlds are created consist of these "absences" or, as Beck puts it at one point, of "second-hand non-experiences" (Beck 1992, pp. 71–72).

This is, in Beck's view, qualitatively new in human history (Beck et al. 1994, p. 3). Obviously, human beings have for millennia traded locally produced goods for goods produced far away, under unknown conditions and by unknown people. But with increasing time–space compression, the known and the unknown, the near and the far away become compressed and impossible to disentangle; the "origins" of the objects that we consume and the contexts in which our actions are inscribed are no longer "places" as much as global flows and networks in their own right. Furthermore, the exact nature of the spatial relationships these "objects" create and rely on can in most cases only be grasped through abstract knowledge, such as statistical forecasts and probability calculations. "[T]he flesh of the world," as Houellebecq puts it in an essay, "is replaced by its digitized image; the being of things is supplanted by the graph of its variations." (Houellebecq 2022, p. 10). This is especially the case when it comes to Beck's primary example: the complex and often global *environmental* impact of human actions. In his view, the erosion of a sense of being contextually situated becomes increasingly pronounced as industrial societies turn toward minimizing the production of often *globally* dispersed "bads"—toxins, radiation, greenhouse gases—rather than maximizing the production of *local* goods (Beck et al. 1994, p. 6).

In their cowritten book, *Reflexive Modernization* (Beck et al. 1994), Beck together with sociologists Anthony Giddens and Scott Lash examine several consequences of this monumental shift for the experience of risk, trust, time, and history. Giddens, for example, focuses on the increasingly important role of *symbolic tokens* (ranging from money to bio- or carbon footprint certifications) and the deterritorialized *expert systems* and the public *trust* that they rely on for their functioning in reflexive modernity. Taken together, these phenomena lead to the dissolution of *tradition* as a general mode of relating to time and space (cf. Giddens 1990, 1991; Beck et al. 1994). Whereas tradition drew its authority from references to a bounded territory and a shared past (real or imagined), reflexivity takes place in a global space and is concerned with projections and forecasts that relate to an uncertain future.

The loss of tradition as way of relating to the world is undoubtedly a fundamental aspect of Houellebecq's work, but here, I want to insist on how reflexivity points to the erosion of the experiential contrast between "near" and "far away," to use Zygmunt Bauman's terminology (Bauman 1998, pp. 13–15). Reflexivity means that what is spatially closest to the individual "reflects" unknown (and even unknowable) global processes, whereas

what is far away takes on a new immediacy or presence through the ever-expanding interconnectedness of the world.

## 8. The World of *Ob-Jects*

It is this experience that the Michel in *Plateforme* (2001) gives voice to as he suddenly becomes aware of the high-technological objects that surround him in an all-inclusive Cuban beach resort: "We lived in a world made up of objects whose manufacture, conditions of possibility [*conditions de possibilité*], and mode of being [*mode d'être*] were completely alien to us" (Houellebecq [2001] 2003, p. 224; 2001b, p. 234).[12] Thrown into the middle of this "world made up of objects" ("In the Middle of the World" is also the subtitle of Houellebecq's tourist novels, *Lanzarote* and *Plateforme*), Michel loses his orientation; the same global modernity that makes the world seem infinitely accessible to the French tourist in Cuba also makes what is *closest* to him (he mentions several objects: a towel, a pair of sunglasses, sunscreen, a swimsuit) radically foreign, as the objects "reflect" global contexts. The massive intrusion of distance destroys the sense of being contextually situated in space, just like in the pornographic scenes analyzed earlier.

Bruno in *Les Particules* makes exactly the same observation as he tries to situate himself in a world of foreign objects:

> All these objects that surround me, that I use or devour, I am incapable of making any of them; I couldn't even begin to understand how they are made. [...] Placed outside of the economic–industrial complex, I couldn't even survive: I wouldn't know how to feed or clothe myself, or protect myself from the weather; my technical competence falls far short of Neanderthal man. (Houellebecq [1998] 2000b, pp. 167–68; 1998b, p. 202. I have slightly altered the translation.)

Again, if Bruno is incapable of understanding the objects that surround him, it is because they are part of reflexively constructed global processes and contexts. Paradoxically, this makes his world smaller, more unknown, and more frightening than that of the Neanderthal. The bringing of the world directly to the *body* of the consumer in the form of objects (the objects that "surround" him, that he "use[s]" and "devours") is a short-circuiting of the world as a space of alterity that can be divided into "near" and "far away." Just like the isolated François in front of his computer screen, Bruno's only "locality" is reduced to his bodily presence, but it is a locality that paradoxically encompasses the entirety of the space of the world. Similarly, in *Plateforme*, it is the bodily presence of the girlfriend Valérie (as well as several Cuban prostitutes) that finally places the narrator and creates what could be called a minimal local context on the Cuban beach, as the cited passage continues: "I slipped two fingers under the bikini; under the artificial fiber construction I could feel the living flesh. [...] This was something I could do, that I knew how to do." (Houellebecq [2001] 2003, p. 225).

Giddens also highlights this *localizing* effect of the body in late modernity. In his account, the body becomes an isolated point with its own inescapable spatio-temporality. But around this minimal locality, the experience of place and time, of being "in the world," becomes increasingly uncertain:

> The constraints of the body ensure that all individuals, at every moment, are contextually situated in time and space. Yet the transformations of place, and the intrusion of distance into local activities [...] radically change what "the world" actually is. (Giddens 1991, p. 187)

Faced with this immediate juxtaposition of the body and the inhuman space of the world, the Houellebecquian protagonists typically take the position of the "banal" consumer who brings the world directly to his own body in acts of detached consumption that turns complexity into the most banal vulgarity.[13] The most concrete manifestation of this "body-world" is undoubtedly the travel agency in *Plateforme*, with its detailed business plan, marketing strategy, and affiliated hotels, all created to transform the Global South into a giant brothel for Western tourists. In fact, the Houellebecquian tourists are, despite their

constant movements across the world, involved in a kind of destruction of space. Their attitude to the space of the world can be summarized, with Laforest's formulation, as a "revulsion" for "the distance that separates [them] from [women's] genitals" (Laforest 2007, pp. 272–73). In turn, the female sexual organs with their, in Houellebecq's work, obsessively described *parois* (vaginal walls) and enveloping tissues and muscles, represent the final form of spatial belonging—as in "Perception–Digestion," where the poet states that "when life has stopped presenting new worlds," only "a sliver of life remains and expires in the dick" (Houellebecq 2014a, p. 320). Or in *Sérotonine*, when Florent-Claude is suddenly stricken by vertigo when he comes to think of all the *chattes* in the world: "billions of pussies on the surface of even a moderate-sized planet, [. . .] it makes you feel dizzy" (Houellebecq [2019] 2020, p. 137).

Such cynical and often pornographic passages ensure that this is the most clearly visible "attitude" toward the world in Houellebecq's work, but there is also a what could be called a "counter discourse" woven into the texts that is related to the structure, system, or the world as a vast, inhuman space. It corresponds to the other side of reflexivity; rather than bringing the world to the body of the subject in the form of detached objects, it points to the dissolution of the subject into the overwhelming space-time of the world.

### 9. Paradoxical Horizons

As I have tried to show, pornography creates a "world" in the ambiguous space between technological deterritorialization and the reterritorialization of sexuality, but it is also associated with a particular "attitude" or way of being in the world. Here, the *ob-jects* are cut off from their immediate contexts and "thrown" in front of the subject, regardless of whether it is the protagonist himself who is in fact decontextualized (as in the case of the sex tourist), or the objects of his desire (as in mass-mediated pornography). This "body-world" of pornography is associated with information, numbers, and data, which can easily be separated from their contexts and recombined in new ways. It is also associated with prose rather than poetry, as we saw in the analysis of the essay "Prise de contrôle sur Numéris"; prose, as Houellebecq often describes it, concerns itself with how "events" intersect in "neutral space and time," whereas poetry abolishes the difference between subject, object, and world altogether (Houellebecq 2022, pp. 38–39).

The "body-world" of pornography is thus a fundamental, structuring element in Houellebecq's work. It is even associated with one of two central characters in several novels: Tisserand in *Extension*, Bruno in *Les Particules*, the narrator in *Lanzarote*, and the *human* Daniel in *La Possibilité* are the most obvious examples. They are the hedonistic consumers who, faced with a loss of locality, strive to bring the world to their bodies in the form of *ob-jects*. But Houellebecq systematically contrasts this "attitude" toward the world with characters that are connected to a different understanding of space: the narrator in *Extension* who, as a programmer at the Ministry of agriculture, encodes physical reality into abstract space; Michel in *Les Particules*, who lives in the global yet subatomic space of quantum physics; the depressed and asexual Rudi in *Lanzarote*; the detached Daniel clone in *La Possibilité*. (Other protagonists, such as Michel in *Plateforme*, Florent-Claude in *Sérotonine*, and François in *Soumission*, veer between the two extremes.) The two schematic sets of characters, which have diametrically opposed ways of relating to the world, also represent different perspectives and voices in the work. Whereas the body-world of pornography is associated with prose and the omnipresence of the body, the counter discourse is related to poetry and the dissolution of the body.[14] At the other side of reflexivity, the limiting, destruction, or loss of the body leads to a negative vision of the world in its entirety. As a conclusion to this article, I will analyze a couple of examples of this less obvious, less brazen counter discourse about the body and the world.

This is the end of *Extension*:

I feel my skin as a frontier, and the external world as a crushing. The impression of separation is total; from now on I am imprisoned within myself. It will not take place, the sublime fusion; the goal of life is missed. It is two in the afternoon.

(Houellebecq [1994] 2011, p. 155; Houellebecq [1994] 2014b, p. 156. I have slightly altered the translation.)

In the final part of the novel, the protagonist goes on a mysterious journey to the source of the Ardèche River that appears as an attempt to unite with the world. Not only is the Ardèche the birthplace of his parents, but it is also the logical endpoint of an increasingly desperate movement from culture to nature, which has taken him from the thoroughly constructed environment of the software company where he works, via rural areas in the Vendée region, to the final destination in the deep forest where the novel ends. On the face of it, the journey is a complete failure. *Spatially*, the narrator is separated from the outside world by his own "skin" that appears as an impenetrable "frontier" ("I feel my skin as a frontier, and the external world as a crushing."). *Temporally*, this final scene takes place on the day of the summer solstice, which contrasts with the protagonist's earlier movements in the novel that were motivated by *cultural* notions of time, such as Christmas and New Year's Eve. Searching for a space of belonging, he is thus confronted with the "crushing" spatio-temporality of the *globe*.

Yet, the passage points to how the loss of experiential place makes the world appear in its absence, as a negation or "separation." The same experience of "separation" that juxtaposes the individual and the world returns, for example, in *Les Particules* when Michel leaves his idyllic childhood home: "He felt *separated from the world* by a vacuum molded to his body like a shell" (Houellebecq [1998] 2000b, p. 72, emphasis added). In *Extension*, Guy de Maupassant is (somewhat surprisingly) singled out as an author who has formulated this new sense of being in the world, as he established "*an absolute separation* between his individual existence and the rest of the world," and the narrator emphasizes that this is the only way in which we can "conceive the world" (*penser le monde*) today (Houellebecq [1994] 2011, p. 147. Emphasis added in the first quote).

In all these examples, it is the world that manifests itself as the direct counterpart to the individual, separated only by the "skin" as a minimal frontier, as the narrator in *Extension* puts it ("I feel my skin as a frontier, and the external world as a crushing."). This metaphor returns in "La peau est un objet limite" ("Skin is a borderline object"). But this time, the poem points explicitly to a painful yet sublime vision of the world in its totality:

> Un moment d'absolue conscience
> Traverse le corps douloureux
> Moment de joie, de pure présence:
> Le monde apparaît à nos yeux. (Houellebecq 2014a, p. 108)

(A moment of absolute consciousness/Passes through the aching body./Moment of joy, of pure presence:/The world appears to our eyes. (Houellebecq 2017, p. 89))

Here, separation as a loss of locality leads to the manifestation of that liminal object the "world," with which the Houellebecquian protagonists are in direct contact. Again, in the state of absolute separation, the world paradoxically appears as the only possible "locality" of the modern individual as the distinctions between near and far away are eroded. In yet another poem, "Dehors il y a la nuit" ("Outside there is the night"), the two final stanzas similarly point to the loss of spatio-temporal locality, in a world "with no human dimension" (Houellebecq 2017, p. 89), and then, immediately, to the possibility of forming a new vision of the world in its entirety. At the end of the poem, the word *monde* (*world*) is repeated three times, like a spell, and the third time, it is emphasized through the breaking of the hexasyllabic rhythm of the poem:

> Nous avons traversé
> Des époques de haine,
> Des temps controversés
> Sans dimension humaine
> Et le monde a pris forme,
> Le monde est apparu

> Dans sa présence nue,
> Le monde. (Houellebecq 2014a, p. 127)

(We have gone through/Ages of hatred,/Controversial times/With no human dimension//And the world has taken shape,/The world has appeared/In its naked presence,/The world. (Houellebecq 2017, p. 223))

In fact, this dual vision of space underpins Houellebecq's work from the very beginning. It is hinted at already in the 1991 essay *Rester vivant*, his advice to a young poet. Like the later novel *Extension*, the essay ends with a view over a landscape that opens up in front of the narrator. "Poetry," the narrator tells us, leads to a "zone of truth": "a different, extremely painful space, but where the view stretches far" (Houellebecq 1998a, p. 26). This is a space, finally, where the dispersed objects are seen in a new light: "the cleared objects appear in their clearness [*les objets nettoyé apparaissent dans leur netteté*], their limpid truth" (p. 26).

But the most thorough manifestation of this "other" space is undoubtedly the ending of *La Possibilité*. In the concluding section of the novel, the clone roams the surface of the destroyed planet Earth trying to reach Lanzarote and the "human presence" of Marie. But just like the journey to the source of the Ardèche River at the end of *Extension*, his excursion is a failure, and he is forced to stop when he reaches the Atlantic Ocean. He decides to lay down in the perfectly still water, where his genetically modified body can absorb the minerals he needs to survive though osmosis. He stays there for the rest of his life, dreaming and observing the ever-changing sky. The final paragraph of the novel describes the clone's failure to reach Lanzarote as a failure to unite himself with the surrounding world. But again, the poetic structure of the passage hints at a "negative" overcoming of the separation between the self and the world.

The entire passage can be read, as Agathe Novak-Lechevalier has pointed out (Novak-Lechevalier 2019, p. 258), as unrhymed alexandrines, a meter with 12 syllables that typically splits up into two sections separated by a caesura. This is not in itself particularly surprising; as Bruno Viard puts it, the reader can often, if he plays around with the pronunciation, find a hexasyllabic rhythm in Houellebecq's prose (Viard 2008, p. 56). But if the passage is read in this way, the caesura in the middle of the "verses" simultaneously splits up and unites the subject and the world in a double movement:

> Le bonheur n'était pas ‖ un horizon possible.
> Le monde avait trahi. ‖ Mon corps m'appartenait
> pour un bref laps de temps; ‖ je n'atteindrais jamais
> l'objectif assigné. ‖ Le futur était vide;
> il était la montagne. ‖ Mes rêves étaient peuplés
> de présences émotives. ‖ J'étais, je n'étais plus.
> La vie était réelle.[15] (Houellebecq [2005] 2012, p. 447)

(Happiness was not ‖ a possible horizon./The world had betrayed. ‖ My body belonged to me/for only a brief lapse of time; ‖ I would never reach/the goal I had been set. ‖ The future was empty;/it was the mountain. ‖ My dreams were populated/with emotional presences. ‖ I was, I was no longer./Life was real. (Houellebecq [2005] 2006, p. 423))

The manifest failure to unite with the surrounding world and the disappearing of the body is charged with a different meaning through the form of the text that unites incompatible elements "in which all possibility of negation is suspended" (Houellebecq 2022, p. 39).[16] In several of these "verses," the subject and the world are confronted on either side of the caesura that marks a transgression or transcendence. Thus, "happiness" in the first verse refers to the subject, and the "horizon" is related to the surrounding world (but it is the "happiness" of belonging to the world, and the "horizon" is a metaphor for the state of mind of the subject). And then, "the world" is immediately juxtaposed with the dying "body" in the second verse, and toward the end, "the mountain," is contrasted with

"my dreams." In this way, the disappearing or death of the subject becomes an opening toward the world through the poetic form of the text. The final verses similarly include oxymoronic contrasts between "brief [...] time" and "never," future "goal" and "empty future," and presence (*présence émotive*) and absence (*je n'étais plus*). And the sequence of contrasts is then suspended in the very last phrase of the novel, "[l]ife was real," since life belongs both to the subject and the world in which it *ek-sists* or reaches beyond itself.

These passages and poems are the exact *inversion* of the "dolly zoom-effect" of pornography, in which the bodily identification of the subject creates a short-circuiting of the world around the object of desire. Here, the loss of the body is the price to pay for a vision of totality that can only be experienced as a negative presence. This faint counter discourse, which is easy to overlook next to the brazen and misogynist voice of banal consumption, does not evoke a world that is more "livable" than the first one. Houellebecq is, after all, an author for whom "[t]he world is suffering unfolded" (Houellebecq 1998a, p. 9). However, these contrasting "reflexive" visions of the world create what could be called, borrowing a notion from Viard (2008, pp. 55–57), a "contrapuntal structure" if they are read together. Without pointing out a clear direction or critique, they create a *destabilization* that makes the ordinary world of banal globalization appear in a new, revealing light. This is also the hope expressed in the multi-media artwork "Opera Bianca" that Houellebecq created with Gilles Touyard and Brice Pauset in 1997. The sung dialogue, which is played over a white space populated by arbitrarily moving abstract shapes, recounts how a world is born. The voices take turns describing how it develops and becomes increasingly incomprehensible as human actions cut it to pieces. But although it is "[i]ndifferent, perfect and round," "the world has kept the memory of its common origin." The countertenor's voice holds out a hope that takes the form of a strange dialectic movement: "Portions of the world appear, then disappear; they appear again." (Houellebecq 2022, p. 69).

**Funding:** This research received no external funding.

**Institutional Review Board Statement:** Not available.

**Informed Consent Statement:** Not available.

**Data Availability Statement:** Not available.

**Conflicts of Interest:** The author declares no conflict of interest.

## Notes

[1]  This formulation originally appeared in two poems in the third section of *La Poursuite du bonheur* (The Pursuit of Happiness, 1991), the prose poem "Le Monde apparaît" (The World Appears) and "La Disparition" (The Disappearance). In both cases, the poems describe people who try to situate themselves in relation to each other in an anonymous cityscape, as in "La Disparition": "We walk in the city, our eyes meet/And this defines our human presence" (Houellebecq 2017, p. 241). The text of this poem was later reused in the part for the soprano in the multimedial artwork "Opera Bianca" (cf. Houellebecq 2022, p. 67). It was also included in Houellebecq's musical album entitled *Présence humaine* (2000).

[2]  See Christophe Ippolito's (2009) discussion of Houellebecq's vision of poetry as a merging with the world that counteracts the separation of subject, object, and outside world that the author sees as inherent in ordinary language. As Ippolite identifies, Houellebecq is deeply inspired by the phenomenological account of poetic language developed by Jean Cohen. This is clearly stated, for example, in the essay "L'absurdité créatrice" ("Creative Absurdity," 1995): "All perception is organized around a twofold difference: between the object and the subject, between the object and the world. [...] Poetry, according to Jean Cohen, leads to a general dissolution of reference points: object, subject, world merge into the same affective and lyrical atmosphere." (Houellebecq 2022, p. 40).

[3]  Sabine van Wesemael has, for example, shown how Houellebecq relies on constantly repeated "trivial" and "standardized" scenarios, such as the recurring *cum-shot* scenes (van Wesemael 2005, p. 191).

[4]  For example, Ben Jeffery (2011) and Lucie Murielle Clément (2018) arguably belong in the first category, Benjamin Boysen (2016) in the second, and Martin Crowley (2002) and Bruno Viard (2008) in the third.

[5]  This is the case, as the title suggests, in many of the photographs displayed at the 2015 Paris exhibition "Before Landing," as well as in several photographs that accompany *Lanzarote* in some editions of the text.

6    As in the *Nicomachean Ethics*: "All art is concerned with the realm of coming-to-be, i.e., with contriving and studying how something which is capable both of being and not being may come into existence [...]." (Aristotle [322] 1962, p. 152).

7    For example, when the narrator returns to Paris in *Extension*: "The arrival in Paris, as grim as ever. The leprous façades, behind which one invariably imagines retired folk agonizing alongside their cat Poucette which is eating up half their pension with its Friskies." (Houellebecq [1994] 2011, p. 82).

8    Beck prefers the term *cosmopolitanism* to *globalization,* because he wants to avoid the economistic connotations that he finds in the latter term (cf. Beck 2006, pp. 8–10). However, the term "globalization" has, in recent years, clearly expanded beyond the economic sphere and become one of the cornerstones of cultural and literary analysis in general. For an overview of the term's history as well as its role in literary studies, see Gupta (2009).

9    As Susan Willis puts it, the supermarket "swaddles the product in First World antiseptic purity and severs its connection with the site of its production. The shopper who enters the air-conditioned supermarket and chooses amongst its papayas, mangos, pineapples, bananas; its winter supply of peaches, nectarines, plums and grapes from Chile, is as unaware of the factors and labor force behind their production as the tourist whose experience of Mexico is an air-conditioned hotel lobby." (Willis 1987, p. 593).

10    One troubling example is the phonetically learned *Shahada*, the Islamic oath that François recites in Arabic at the end of *Soumission* (Houellebecq [2015] 2016, p. 248).

11    All of these terms are associated with different scholars and theories about globalization, postcolonialism, and culture. Beck discusses some of these tropes in *Cosmopolitan Vision* (Beck 2006, pp. 31–32). For a more general discussion of these terms, see also Burke (2009).

12    I have slightly altered the translation.

13    But at the same time, the Houellebecquian protagonists are chronically unsure whether the world they live in is too complex or too simple, for example: "the world around me [...] had become complex, and now [...] I could no longer deal with the complexity of the world into which I had been plunged [...]." (Houellebecq [2019] 2020, p. 255); "we live in such a simple world [...]. [...] Is it really possible to live and to believe that there's nothing else?" (Houellebecq [1994] 2011, p. 147); "It's not me who's strange, it's the world around me." (Houellebecq [2001] 2003, p. 328). In an insightful analysis, Delphine Grass similarly underscores how modern, functional architecture, such as the business center La Défense, and modern communication technologies share a fixation with transparency that paradoxically makes society even more impenetrable or opaque in Houellebecq's work: "Transparency is not experienced by the workers of La Défense as 'transparency.' On the contrary, Houellebecq argues, the process of material production has never been more opaque to those workers. [...] [T]he visual transparency and apparent realism of modern architecture creates a linguistic illusion, a communication decoy for the city's workers who are tempted to associate the transparency of the offices with more freedom and a greater space of interaction." (Grass 2011, p. 342).

14    In his important study of Houellebecq's poetry, David Evans (2007) similarly highlights a recurrent opposition between what he calls the "self" and the "structure." The "structure" represents a protection from the passions, pain, and sexuality that is associated with individual existence—in his reading of one of the poems, the reader must even actively choose between eliding the word *I* (*je*) or sacrificing the meter, or the "structure," of the poem (cf. Evans 2007, p. 209).

15    My way of structuring the "verses" in this passage differs slightly from Novak-Lechevalier's (cf. Novak-Lechevalier 2019, p. 258).

16    Compare Houellebecq's description of poetic language in the essay on Jean Cohen's *La Structure du langage poétique*: "Poetry [...] aims to produce a fundamentally alogical discourse, in which all possibility of negation is suspended. [...] Poetic deviations [...] aim to create an 'effect of limitlessness' where the field of affirmation invades the whole world, without letting the contradiction subsist as an outside." (Houellebecq 2022, p. 39).

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
