# Peer review of "Planet YouPorn: Pornography, Worlding, and Banal Globalization in Michel Houellebecq’s Work"

_humanities, doi:10.3390/h12020031_

Round 1

Reviewer 1 Report

This is a strong and original article that offers detailed and astute analysis of neglected areas of Houellebecq’s work. The article covers so much ground, however, that at times the direction of the argument can be a little difficult to follow. I would recommend having a stronger statement of intention at the outset with clearer signposting throughout.

To that end, the application of Beck’s concept of ‘worlding’ to Houellebecq’s work is original, but the sense and pertinence of this concept needed more carefully setting out in the introduction.

The spatial language surrounding Houellebecq’s discourse of love and sex is well observed, ably supported by a good number of quotations and the metaphor of the ‘dolly-zoom effect’ captures very neatly the novelist’s paradoxical relation to the form. Likewise, the ubiquity of images of ‘falling into the world,’ in Houellebecq’s work are helpfully pointed out.

The reading of ‘Prise de contrôle sur Numéris,’ a rarely discussed early work by Houellebecq, is very welcome and helps to demonstrate a consistency of theme and approach across the author’s work.

The article does an impressive job of sampling judiciously within the vast secondary literature on Houellebecq.

In summary, again: a valuable and original contribution but that would benefit from a clearer statement of purpose in the introduction.

Reviewer 2 Report

A text whose main subject, pornography, is at the heart of the author's burning news. It even goes beyond fiction since Houellebecq recently shot in the pornographic film “Kirac 27” by Dutch director Stefan Ruitenbeek. It would seem that since then, the writer has decided to backtrack and wishes to ban the film's release, scheduled for March 11. To be continued…

This anecdote reinforces all the more the interest in this manuscript with an original subject that explores the notions of distance and relationship to the world of the individual through the study of different passages taken from the works of Michel Houellebecq, novels, and poems. It presents Houellebecq's relationship to globalization in a new light, notably with the use of “tools” such as the “dolly zoom”.

The author of the manuscript demonstrates an in-depth knowledge of Houellebecq's works and masters the concepts he uses in his study (“banal cosmopolitanism”, “reflexivity”). The subject is clearly stated and well-structured. Interesting ideas but nevertheless quite complex required a such organization to orient the reader. The underlying theory (Beck, Giddens) is very relevant and illuminates the text. The various references and contextualization are also appropriate.

I found one or two typos that should be corrected:

145: “La Possibilié” instead of “La Possibilité” (in the title #3) – “t” is missing

622: a “L” is missing in Houellebecq’s name

And a repetition:

648: “it is the world that manifests itself as the direct counterpart to the individual”

and after quotation,

662: “The world manifests itself as the direct counterpart to the individual”.

These remarks are minor, and I enjoyed reading this manuscript which I recommend for publication in Humanities.

Reviewer 3 Report

This article has much to recommend it, particularly in terms of the close textual readings of Houellebecq’s work through the lens of Beck’s theories of ‘banal cosmopolitanism’ and ‘reflexivity’. The existing scholarly context for the author’s focus is well set out, particularly in relation to the question of ‘being in the world’. The argument centres around love and its technological, globalised, mediatised counterpart – pornography – as a means of considering the experience of being in the world. Sex, for example, is seen as a means of transcending the self and merging with the space of the world, for it is – the author argues – in the space of alterity between self and world, in the copresence of distance and presence that the human world is created, and poetry is a worlding force. However, the forces of modernity and globalization – here represented by pornography – complicate this, and the relationship of the body to space changes in the technological world.

Within this broad conceptual frame, there are many interesting and valuable analyses. The close textual readings of the texts are very well done, and the framework of banal cosmopolitanisms is particularly well deployed. I would suggest, however, that more work is done across the article to signpost the key argument throughout, as I lost sight of this on a number of occasions which made the overall article harder to follow. For example, the discussion of ‘falling’ into the human and even the push/pull dolly zoom analysis were very interesting but their place within the overall argument could be more explicitly signposted. This might simply be a question of summing up each of the subsections with a link to the overarching argument. 

The prose/poetry distinction is also clearly very important for the argument. The author notes the following on p.4:

‘At the end of the article, finally, I outline a counter discourse in Houellebecq’s work that reverses the relationship between body and world that is associated with and promoted by pornography. Rather than prose, it is associated with poetry, and rather than bringing the world to the body of the subject, it points to the dissolution of the subject into the space of the world. These two voices or discourses mirror each other, and they are, I argue, complimentary responses to the experience of banal globalization.’

However, this doesn’t come across strongly in the ensuing discussion. A more developed conclusion may address this issue.

In short, while I am not a Houellebecq specialist, I see significant value in the discussion, and would note that it also has broader interest beyond Houellebecq studies. Above all, therefore, my suggestions relate to framing, signposting, and clarifying the overall argument.
